# Integrated Analysis of the Transcriptome and Microbial Diversity in the Intestine of Miniature Pig Obesity Model

**DOI:** 10.3390/microorganisms12020369

**Published:** 2024-02-10

**Authors:** Wenjing Qi, Siran Zhu, Lingli Feng, Jinning Liang, Xiaoping Guo, Feng Cheng, Yafen Guo, Ganqiu Lan, Jing Liang

**Affiliations:** 1College of Animal Science and Technology, Guangxi University, Nanning 530004, China; 1818401004@st.gxu.edu.cn (W.Q.); gqlan@gxu.edu.cn (G.L.); 2Laboratory Animal Center, Guangxi Medical University, Nanning 530021, China

**Keywords:** obesity model, Guangxi Bama miniature pig, transcriptome, gut microbiota, immunization, gene–microbe interactions

## Abstract

Obesity, a key contributor to metabolic disorders, necessitates an in-depth understanding of its pathogenesis and prerequisites for prevention. Guangxi Bama miniature pig (GBM) offers an apt model for obesity-related studies. In this research, we used transcriptomics and 16S rRNA gene sequencing to discern the differentially expressed genes (DEGs) within intestinal (jejunum, ileum, and colon) tissues and variations in microbial communities in intestinal contents of GBM subjected to normal diets (ND) and high-fat, high-carbohydrate diets (HFHCD). After a feeding duration of 26 weeks, the HFHCD-fed experimental group demonstrated notable increases in backfat thickness, BMI, abnormal blood glucose metabolism, and blood lipid levels alongside the escalated serum expression of pro-inflammatory factors and a marked decline in intestinal health status when compared to the ND group. Transcriptomic analysis revealed a total of 1669 DEGs, of which 27 had similar differences in three intestinal segments across different groups, including five immune related genes: *COL6A6*, *CYP1A1*, *EIF2AK2*, *NMI*, and *LGALS3B*. Further, we found significant changes in the microbiota composition, with a significant decrease in beneficial bacterial populations within the HFHCD group. Finally, the results of integrated analysis of microbial diversity with transcriptomics show a positive link between certain microbial abundance (*Solibacillus*, *norank_f__Saccharimonadaceae*, *Candidatus_Saccharimonas*, and *unclassified_f__Butyricicoccaceae*) and changes in gene expression (*COL6A6* and *NMI*). Overall, HFHCD appears to co-contribute to the initiation and progression of obesity in GBM by aggravating inflammatory responses, disrupting immune homeostasis, and creating imbalances in intestinal flora.

## 1. Introduction

Obesity is a significant risk factor for various chronic diseases, including diabetes, hypertension, and coronary heart disease, and has become a major public health problem worldwide [1]. Currently, the exact causes of obesity are not fully clear, but it is believed to be related to factors such as diet, exercise, genetic factors, dysbiosis of the gut microbiota, and inflammatory responses [2]. Pigs can spontaneously develop obesity, and they are not only similar to humans in terms of the anatomy and physiology of the cardiovascular and digestive systems but also exhibit a high degree of similarity in the process of glucose and lipid metabolism. Meanwhile, due to being omnivorous animals, pigs are highly suitable for studying diet-related models and translating research findings to humans [3]. In particular, the Guangxi Bama miniature pig (GBM) possesses characteristics such as small body size, ease of rearing, strong adaptability, and genetic stability, making them an ideal experimental animal model for studying diet-related cardiovascular metabolic diseases [4,5,6]. GBM is a breeding line that has been inbred for over 30 years (since 1987), featuring a comparatively stable genetic background [4,7]. This makes it a critical animal model for researching human diseases. However, in obesity research, there is a greater prevalence of using mouse models, while the utilization of pigs is relatively limited. Therefore, it is of significant importance to establish an obesity model using GBM.

Transcriptomics, as a sequencing method, studies the transcription of all genes and their transcriptional regulation patterns of organisms in a specific physiological process at the mRNA expression level. It can be used to predict the role of individual genes activated in response to external stimuli [8]. Therefore, transcriptomics has been widely applied in the search for therapeutic targets for diseases. Meanwhile, the changes in intestinal bacterial communities affect the pathogenesis of the disease [9]. There exists a complex association between gut microbiota and obesity [10]. Previous studies have found that significant changes occur not only in mRNA expression levels [8,11], but also in the abundance of gut microbiota [12,13] in obese humans and mice, reaching statistically significant levels.

This study utilized GBM fed with high-fat, high-carbohydrate diets (HFHCD) as an obesity model. Transcriptomics and microbial diversity analysis techniques were applied to analyze the jejunum, ileum, and colon segments. The aim of the study was to identify potential biomarkers associated with obesity and provide reference for the use of GBM in human obesity research and applications.

## 2. Materials and Methods

### 2.1. Establishment of the Obesity Model

Ten female GBM weighing between 17–20 kg from the Guangxi Bama Miniature Pig Breeding Center were randomly divided into two groups (*n* = 5 per group), including the normal diet (ND) group and the HFHCD group. The ND group was fed a normal diet throughout the experiment, while the HFHCD group was fed with 60% standard diet, 30% sucrose, and 10% soybean oil, twice daily, with free access to water. The experimental period lasted for 26 weeks. The standard of nutritional ingredients of the miniature pig feed are shown in Table 1.

### 2.2. Material Collection

Jejunum, ileum, and colon tissues and their contents were collected. They were used for transcriptome sequencing, microbial diversity analysis, and preparation of pathological slides.

### 2.3. Measurement of Body Parameters

Body weight (BW), body length, abdominal circumference (AC), and backfat thickness (BT) of all pigs were collected, and body mass index (BMI) was calculated. The formula for calculating *BMI* of pigs is as follow [14]:$$BMI=\frac{Body\;weight\;\left(\mathrm{kg}\right)}{Body\;length\;{\left(m\right)}^{2}}$$

### 2.4. Measurement of Blood Physiological and Biochemical Indicators

Blood samples were collected from the anterior vena cava of all pigs after 16 h of fasting. The collected blood was centrifuged at 4 °C and 2500 rpm for 15 min to obtain serum. Colorimetry was utilized to determine the levels of fasting blood glucose (*FBG*), triglycerides (TG), total cholesterol (TC), high-density lipoprotein (HDL), low-density lipoprotein (LDL), and free fatty acids (FFA) in the serum. Additionally, ELISA kits (Thermo Fisher Scientific, Waltham, MA, USA) were employed to measure the levels of fasting insulin (*FINS*), interleukin-6 (IL-6), and tumor necrosis factor-alpha (TNF-α). The homeostasis model assessment of insulin resistance (*HOMA-IR*) was calculated using the following formula [15]:$$HOMA-IR=\frac{FBG\;\left(\frac{\mathrm{mmol}}{L}\right)\;\times \;FINS\;\left(\mathrm{mIU}/L\right)}{22.5}$$

### 2.5. Assessment of Tissue Pathology

The pigs were euthanized and subsequently dissected under sodium pentobarbital (Sigma-Aldrich, St. Louis, MO, USA) anesthesia. Segments of the jejunum, ileum, and colon were collected and fixed in 4% neutral formalin (Sigma-Aldrich, St. Louis, MO, USA). Next, routine paraffin sections were made and stained with hepatic hematoxylin and eosin (H&E) to observe the pathological changes under an electron microscope and analyze them.

### 2.6. Transcriptome Sequencing

Three experimental pigs were randomly selected from both the ND group and the HFHCD group. Tissue samples from the jejunum, ileum, and colon were collected for transcriptomic sequencing analysis. The comprehensive methodologies for mRNA library construction and sequencing, data analysis, enrichment analysis of DEGs consist of screening of DEGs, GO, KEGG, and GSEA are detailed in the Appendix A [16,17,18].

### 2.7. Validation Using qRT-PCR

The RNA-seq results were validated through qRT-PCR, with the detailed steps and procedures available in the Appendix A. Primer sequences are also provided in the Appendix A.

### 2.8. Microbial Diversity Analysis

The contents of the intestine, including the jejunum, ileum, and colon, were collected from the ND group and HFHCD group (*n* = 5 per group) for 16S rRNA sequencing of the intestinal microbiota. The comprehensive methodologies for total DNA library preparation, sequencing, ASVs classification, alpha diversity, community composition, and differential comparison are detailed in the Appendix A [19,20,21,22,23,24].

### 2.9. Integrated Analysis of Microbial Diversity with Transcriptomics and Physiological Biochemical Indicators

The correlation between common DEGs in three intestinal segments—*COL6A6*, *CYP1A1*, *EIF2AK2*, *NMI*, and *LGALS3B*—and the microbiota at various taxonomic levels (phylum, class, order, family, genus) was determined using Spearman’s algorithm. The selection criteria included a correlation coefficient’s absolute value > 0.5 and *p* adjust < 0.05. This analysis facilitated the visualization of the mRNA–microbiota network in Cytoscape software (version 3.3.0). Further, Spearman’s algorithm enabled the examination of correlations between intestinal microbiota and physiological and biochemical indicators, employing the rcorr function for calculating correlation coefficients and *p*-values, with a threshold of *p* < 0.05. The relationship between intestinal microbiota and physiological and biochemical indicators was visually represented using the corrplot function.

### 2.10. Statistical Analysis

Statistical software SPSS 22.0 was used for data analysis. All measurement data were expressed as mean ± standard deviation $(\bar{x}\;\pm \;s,\;n=3/5$). Two groups of data were analyzed by Student’s t test and Wilcoxon rank-sum test. Represented by *p* < 0.05 means the differences were statistically significant.

## 3. Results

### 3.1. HFHCD Induced an Elevation in the Obesity Index

The BW, AC, and BT were measured, and the BMI was calculated to investigate the impacts of HFHCD on the occurrence of obesity in GBM. The results suggest that these parameters were significantly enhanced in the HFHCD group compared with the ND group (*p* < 0.05) (Figure 1A,B).

### 3.2. HFHCD-Induced Dyslipidemia, Hyperglycemia, and Inflammatory Response

The levels of FBG, FINS, TG, TC, FFA, IL-6, and TNF-α were determined, and the HOMA-IR index was calculated to explore the effects of HFHCD on blood indicators in GBM. The results reveal that the indicators of blood glucose metabolism, FBG, FINS, and HOMA-IR (Figure 1C), as well as the blood-lipid-related indicators TG, TC, and FFA (Figure 1D) and the inflammatory factors IL-6 and TNF-α (Figure 1E), were significantly increased in the HFHCD group compared with the ND group (*p* < 0.05).

### 3.3. Obesity-Impaired Gut Health

H&E staining of the paraffin sections of the intestinal tissues was conducted to evaluate the health status of the intestines. The structure of the jejunum, ileum, and colon with well-arranged villi and intact crypts and no significant histological abnormalities were observed in the ND group. In comparison, the height and density of the villi and the number of goblet cells was reduced in the HFHCD group. Additionally, damage to the intestinal mucosa and villous atrophy were observed in the HFHCD group (Figure 1F). The ratio of villi height to crypt depth (V/C ratio) in the jejunum and ileum and the crypt depth in the colon were significantly reduced in the HFHCD group (*p* < 0.05) (Figure 1G).

### 3.4. Transcriptomics and Validation

#### 3.4.1. Identification of Differentially Expressed Genes

RNA-seq was used to screen the DEGs among the jejunum, ileum, and colon to analyze further the molecular events in the obesity model of GBM. In total, 1669 DEGs were identified in the HFHCD/ND groups, including 398 in the jejunum, 834 in the ileum, and 437 in the colon (Figure 2A,B). The results from the enrichment analysis of DEGs reveal that, across all three intestinal segments, there is significant consistency within each group and distinct differentiation between the two groups (Figure 2C).

Functional enrichment using GO, KEGG, and GSEA was performed to determine the functions of the DEGs. The 15 topmost results from the GO enrichment analysis were selected (Figure 2D). The results suggest that the GO terms common to the three intestinal segments are mainly enriched in processes such as immune system processes, extracellular region, defense response, immune response, response to external stimulus, response to biotic stimulus, and response to other organisms. The results of the KEGG enrichment analysis (Figure 2E) reveal nine enriched pathways in the jejunum, including the AGE–RAGE signaling pathway in diabetic complications, PI3K–Akt signaling pathway, ECM–receptor interaction, and viral protein interaction with cytokine and cytokine receptors. In the ileum, 29 differentially enriched pathways were identified, of which 14 belonged to the second category of the immune system and immune disease, including the intestinal immune network for IgA production, B cell receptor signaling pathway, and inflammatory bowel disease. In the colon, 11 differentially enriched pathways were identified, with 8 pathways belonging to the second category of the immune system and immune disease, including the intestinal immune network for IgA production, B cell receptor signaling pathway, and antigen processing and presentation. The results of the GSEA enrichment analysis demonstrate that six common pathways were activated among the three comparison groups (Figure 2F), including two (DNA replication and systemic lupus erythematosus) that were up-regulated and four (focal adhesion, hypertrophic cardiomyopathy (HCM), ECM–receptor interaction, and protein digestion and absorption) that were down-regulated.

#### 3.4.2. Quantitative RT-PCR Verification

Among the total DEGs, 27 of them were common to the three different intestinal segments with consistent expression trends, 20 were up-regulated, and 7 were down-regulated. qRT-PCR was used to examine the expression levels of the 5 DEGs from the 27 common DEGs associated with inflammation- and immune-associated responses and validate the results of the RNA-seq analysis. The results demonstrate a significant down-regulation of *COL6A6* and *CYP1A1*, while *EIF2AK2*, *NMI*, and *LGALS3B* are significantly up-regulated (*p* < 0.05) (Figure 2G). These findings are consistent with the transcriptome data, further supporting the high quality of the RNA-seq results.

### 3.5. Microbial Diversity Analysis

#### 3.5.1. Sample Sequence Information

To investigate the changes in gut microbiota within an obesity model of GBM, we sequenced 30 samples of jejunum, ileum, and colon contents from 10 pigs in ND and HFHCD groups using high-throughput sequencing targeting the 16S rRNA v3–v4 region. After denoising and optimizing the sequences obtained from quality-controlled splicing, a total of 5305 ASVs were acquired. Subsequent rarefaction analysis, established on a minimum sample sequence count of 17,449, yielded 2246 ASVs. The classification of these ASVs revealed the presence of 27 phyla, 56 classes, 122 orders, 209 families, 429 genera, and 644 species.

#### 3.5.2. Diversity Analysis

To assess the changes in microbial community diversity, we calculated the alpha diversity indices and performed inter-group difference analysis. Our analysis of the Sobs, ACE, Chao, Shannon, and Simpson indices reveals two significant differences: in the HFHCD group, the jejunum’s Shannon index was significantly higher, and the Simpson index was significantly lower compared with the ND group (Figure 3A). In both the ileum and colon, the diversity indices of the intestinal microbiota exhibited no significant variations between those in the ND and FHHCD groups.

To investigate microbial community composition and structural differences between the ND and HFHCD groups, principal co-ordinates analysis (PCoA) was conducted to assess the similarity of microbial community structures across samples. Figure 3B depicted the explanatory power of PCoA, ranging from 7.71% to 58.59% for various sample groups. The PCoA sample distribution revealed an overlap in the ASV-level microbial composition between the jejunum and ileum of both groups, suggesting limited differentiation. Conversely, colon samples displayed complete separation by group, indicating a higher degree of variability. These results indicate that obesity significantly affects the intestinal microbiota structure of GBM, with a more pronounced effect on the colon than on the jejunum and ileum.

#### 3.5.3. Community Composition and Variation

At the phylum level, 27 phyla were identified in both the ND and HFHCD groups. Of these, 17 phyla were common to both groups in the jejunum, 12 in the ileum, and 17 in the colon. Firmicutes was the predominant phylum in both the jejunum and ileum for both groups, while in the colon, Firmicutes and Bacteroidota were dominant. No significant differences were observed in the species composition at the phylum level across the three intestinal segments (Figure 3C).

At the genus level, a total of 429 bacterial taxa were identified. Both the ND and HFHCD groups shared 284 genera in the jejunum, with 11 unique to the ND group (Appendix A) and 223 unique to the HFHCD group (Appendix A). Lactobacillus and Bacillus were the predominant genera in both groups (Figure 3D). In the ileum, 182 genera were common between both groups, alongside 108 unique to the ND group (Appendix A) and 21 unique to the HFHCD group (Appendix A). These genera were again dominated by Lactobacillus and Bacillus (Figure 3D). Within the colon, the groups shared 197 genera, with 20 unique to the ND group (Appendix A) and 47 unique to the HFHCD group (Appendix A). Comparatively, the HFHCD group showed a marked decrease in the relative abundance of *Treponema*, *Lachnospiraceae_XPB1014_group*, *Eubacterium_ruminantium_group*, *Candidatus_Saccharimonas*, and *Solibacillus*, while showing an increase in *unclassified_f__Butyricicoccaceae* (*p* < 0.05). Interestingly, Lactobacillus exhibited a decreased relative abundance in all three intestinal segments within the HFHCD group, although these changes were not statistically significant (Figure 3E).

#### 3.5.4. Differences in Species Composition

Utilizing linear discriminant analysis and effect size (LEfSe) analysis, significant disparities in bacterial taxa abundance were identified between the ND and HFHCD groups. This analysis aimed to pinpoint species characteristics that most clearly differentiate these groups and assess their impacts. Comprehensive differential analysis across five taxonomic levels—phylum, class, order, family, and genus—identified 13 significant differences, all located in the colon. The LDA results highlight 10 significant disparities within the ND group, including *g__Treponema*, o__Spirochaetales, *p*__Spirochaetota, f__Spirochaetaceae, c__Spirochaetia, *g__Lachnospiraceae_XPB1014_group*, *g__Solibacillus*, *g__Eubacterium_ruminantium_group*, *g__Candidatus_Saccharimonas*, and f__Saccharimonadaceae. In contrast, the HFHCD group exhibited three significant differences, specifically in f__Butyricicoccaceae, *g__Dorea*, and *g__unclassified_f__Butyricicoccaceae* (Figure 3F).

### 3.6. Integrated Analysis of Microbial Diversity with Transcriptomics and Physiological Biochemical Indicators

Utilizing RNA-seq analysis, inflammation-related DEGs common to three segments of the intestine were identified: *COL6A6*, *CYP1A1*, *EIF2AK2*, *NMI*, and *LGALS3B*. Then the correlations between these DEGs and the gut microbiota were investigated using Spearman correlation analysis (Figure 4A). The findings demonstrated a positive correlation between the expression fluctuations of COL6A6 and the abundance alterations of *Solibacillus* (r = 0.56, *p* adjust = 0.02), *norank_f__Saccharimonadaceae* (r = 0.53, *p* adjust = 0.03), and *Candidatus_Saccharimonas* (r = 0.51, *p* adjust = 0.04). Notably, the expression changes in NMI also positively correlated with the abundance changes in un*classified_f__Butyricicoccaceae* (r = 0.55, *p* adjust = 0.03). Furthermore, the correlation between these intestinal microbiota and physiological and biochemical indicators was calculated using Spearman correlation analysis. The results revealed significant correlations between specific microbiota—*Solibacillus*, *norank_f__Saccharimonadaceae*, *Candidatus_Saccharimonas*, and *unclassified_f__Butyricicoccaceae*—and key physiological and biochemical indicators. Notably, *Solibacillus*, *norank_f__Saccharimonadaceae*, and *Candidatus_Saccharimonas* exhibited negative correlations with TG, TC, FFA, and IL-6. Conversely, *unclassified_f__Butyricicoccaceae* demonstrates a positive correlation with these metrics (Figure 4B).

## 4. Discussion

Poor dietary habits are a key factor in causing obesity, which emphasizes the fact that healthy eating habits are particularly necessary. Compared with mice, pigs serve as superior models for being genetically and physiologically more similar to humans [3]. GBM, characterized by its diminutive stature, ease of breeding, robust adaptability, and genetic stability, has emerged as an ideal subject for investigating diet-related cardiovascular metabolic diseases. Our preceding studies have established GBM as one of the optimal candidate for developing a type 2 diabetes (T2DM) model, yet the underlying molecular mechanisms remain elusive [4,25]. The utilization of female miniature pig models in the studies of metabolic diseases has yielded gratifying outcomes [26,27]. This study aims at creating a reliable obesity model and combines transcriptomic analysis and microbial diversity studies to clarify the impact of obesity on GBM. It offers valuable insights into the mechanisms driving the onset and progression of HFHCD-induced GBM obesity.

BMI serves as a significant metric that mirrors weight characteristics and aids in evaluating the obesity index [14]. Our study reveals that pigs subjected to the HFHCD exhibited significant increases in BW, AC, and BT compared with the normal diet (*p* < 0.05). This finding suggests that HFHCD induced obesity in GBM. Furthermore, FBG, FINS, and HOMA-IR serve as indicators of glucose metabolism, while TG, TC, and FFA are associated with blood lipids [28]. IL-6 and TNF-α are cytokines utilized for evaluating the level of inflammation in the body [29]. The findings of this study indicate that the GBM that consumed a HFHCD exhibited aberrations in markers for glucose, lipids, and inflammation, concurrent with early insulin resistance. These findings imply that HFHCD can trigger metabolic disorders related to glucose and lipid, inflammation, and insulin resistance in GBM. Key indicators of intestinal digestion and absorption considered includes villus height, density, and crypt depth on the mesentery of the small intestine. The quantity of goblet cells also serves as a crucial gauge of intestinal health. The V/C provides a comprehensive overview of the nutrient absorption capacity and health of the small intestinal villi [30]. The depth of the colon crypt denotes its overall functionality [31]. Major pathological changes observed in the HFHCD group’s intestinal tissue encompassed a reduction in villus height and density, a decrease in goblet cells, damage to the intestinal mucosa, and villus deformation. Concurrently, there was a significant decrease in the villi of the jejunum and ileum V/C value, and a significant diminution in colon crypt depth (*p* < 0.05). These findings suggest that HFHCD can compromise the digestive and absorptive capacities of the GBM intestine, generally undermining intestinal health. which are analogous to the pathological alterations seen in human obesity. Consequently, we have successfully established a GBM obesity model.

Obesity, a chronic metabolic disorder, arises from abnormal energy metabolism resulting in excessive fat accumulation. Immune and inflammatory responses play a crucial role in the development of obesity [32]. The GO enrichment analysis reveals seven common terms with significant differences across the jejunum, ileum, and colon. Importantly, all these terms bear a close relationship with immune and inflammatory responses. Furthermore, in the jejunum, KEGG analysis shows differential enrichment in pathways associated with inflammation, immunity, and metabolic disorders such as diabetes. In the ileum, 14 of the 29 differentially enriched KEGG items pertain to the immune system and immune disease. Similarly, in the colon, 8 out of 11 differential KEGG items target the immune system and immune disease. GSEA results identify six common activated pathways across these intestinal segments, with half relating to immunity and inflammation. Collectively, GO, KEGG, and GSEA analyses converge on immune and inflammatory responses, hinting at a link between GBM obesity’s emergence and alterations in the immune system.

*COL6A6* decreases pro-inflammatory cytokines (MCP-1, IL-1β) and elevates Th2 cytokines (IL-4, IL-10) in mouse serum [33]. *CYP1A1*, targeted by the aromatic hydrocarbon receptor (AhR)—a nuclear receptor modulating intestinal inflammation—exhibits reduced expression during inflammatory states [34]. *EIF2AK2* selectively down-regulates transcription of innate immune response genes [35]. *NMI* promotes sepsis inflammation by activating the NF-κB pathway and releasing pro-inflammatory cytokines [36]. *LGALS3BP*, a galectin ligand, triggers a pro-inflammatory pathway [37]. They are all implicated in immune responses. In our research, the HFHCD group demonstrated significant down-regulation of *COL6A6* and *CYP1A1* and up-regulation of *EIF2AK2*, *NMI*, and *LGALS3BP* across the three intestinal segments. The alterations in the expression levels of these genes may contribute to the initiation of the body’s inflammatory response. Nevertheless, as of the present moment, no substantiating evidence has been unearthed to indicate their direct implication in the onset and progression of obesity. Consequently, these genes stand as potential pivotal candidates for HFHCD-induced obesity. Furthermore, serum testing results show a significant increase in the levels of pro-inflammatory cytokines IL-6 and TNF-α. In conclusion, HFHCD may affect the occurrence and progression of obesity by influencing inflammation and immune balance of GBM.

As the “invisible organ” and harboring the human body’s “second genome”, the gut microbiota plays a pivotal role in host energy metabolism, immune function, and inflammatory responses. This impact extends to conditions like obesity and insulin resistance [38,39]. Dietary choices, particularly HFHCD, can swiftly modify the gut microbiota composition, thereby contributing to obesity development [40]. With evolving research, the mechanisms through which specific gut microbiota regulate obesity are being incrementally unveiled. For example, *Akkermansia muciniphila* has demonstrated potential in mitigating metabolic disorders induced by high-fat diets, such as escalated fat mass and insulin resistance, albeit to a certain extent [41,42]. *Clostridium coccoides*, a bacterium recognized for its fat reduction capabilities, can decrease fat content when its abundance is increased [43,44]. These alterations in gut microbiota play a crucial role in the onset of obesity. Additionally, emerging research highlights a profound interconnection between gut microbiota dysbiosis and intestinal inflammation. Under normal conditions, there exists a symbiotic equilibrium between the gut microbiota and its host. This harmony is pivotal in maintaining the integrity of the intestinal mucosa, modulating the immune system, and generating metabolites that are beneficial for the host. However, when this delicate balance is disturbed, it may result in an increase in pathogenic bacteria and a decrease in beneficial ones. Such an imbalance can initiate an immune response in the intestinal mucosa, leading to inflammation. This inflammatory response can further compromise the intestinal mucosal barrier, thereby intensifying the dysbiosis [45]. Moreover, the intestinal inflammation itself can catalyze changes in the gut environment, influencing the microbiota composition. The mucosal damage and altered immune responses brought on by inflammation can foster an environment that favors certain pathogens, while being hostile to beneficial bacteria, thus, exacerbating the dysbiosis [46]. Consequently, the dynamics between gut microbiota dysbiosis and intestinal inflammation are intertwined, potentially spiraling into a deleterious cycle: microbiota dysbiosis triggers intestinal inflammation, which, in turn, exacerbates the disruption of the microbiota equilibrium.

The Shannon index is directly proportional to biodiversity, while the Simpson index is inversely proportional to biodiversity [47]. While some studies have demonstrated a decrease in the diversity of the gut microbiome in individuals suffering from obesity and conditions like T2DM [48,49], a number of studies present contrary evidence, challenging this widely accepted trend. For example, this research indicates that the diversity of the gut microbiome in individuals with obesity is notably greater compared to those who are not obese [50]. In comparison to non-diabetic individuals, those diagnosed with diabetes demonstrate enhanced diversity in the microbiome of their fecal samples [51]. Our study revealed that in the jejunum of the HFHCD group, the Shannon index was elevated, and the Simpson index was reduced. This indicated an elevated level of alpha diversity, which is consistent with some of the results documented in the previously referenced studies. This finding corresponds with the identification of a greater variety of specific bacterial genera in the jejunum. Notably, this group included harmful bacteria like Veillonella, capable of colonizing the intestines under inflammatory conditions [52]. The escalation in the presence of microbes implicated in inflammatory processes might have contributed to the augmented microbial diversity. The dynamics of immune and inflammatory responses emerge as critical areas of focus in obesity model research, underscoring the potential outcome of a reciprocal interplay between the host and gut microbiota during the development and progression of obesity. To sum up, when it comes to exploring the diversity of the gut microbiome in individuals with obesity, it is imperative that further detailed investigations are conducted to uncover the intricate mechanisms at play. Conversely, inapparent differences in the alpha diversity in the ileum and colon contents show the pronounced influence of obesity on the alpha diversity of the intestinal microbiota in the jejunum of GBM, and lesser effects on the ileum and colon. In the PCoA analysis, the ND group and HFHCD group show some overlap in the microbial composition at the ASV level in the jejunum and ileum, with a lower degree of separation, whereas the colon samples were completely separated in different groups, indicating a higher degree of dispersion. These findings suggest that, in comparison to healthy pigs, the obesity model of GBM exhibits smaller variations in the microbial community structures of the jejunum and ileum, yet substantial differences in that of the colon.

In this study, Firmicutes emerged as the dominant phylum across all three intestinal segments, accompanied by the minor presence of Bacteroidota and Actinobacteriota in certain groups, which aligned closely with established research [53]. Notably, the species composition at the phylum level showed no marked differences across these segments, suggesting a minimal impact of obesity on the phylum-level microbiota in the intestines of GBM. The *Lachnospiraceae_XPB1014_group* exhibited a specific negative correlation with Crohn’s disease, a form of inflammatory bowel disease [54]. The *Eubacterium_ruminantium_group* has been shown to alleviate colitis [55]. *Solibacillus*, part of the Bacillus genus, has an as-yet undetermined function. Earlier research indicated a significant reduction in *Solibacillus* during the initial stages of colorectal mucosal carcinogenesis [56]. Our study did not identify significant genus-level differences in the jejunum and ileum across the two groups. However, the HFHCD group demonstrated a significant decrease in the relative abundance of *Lachnospiraceae_XPB1014_group*, *Eubacterium_ruminantium_group*, and *Solibacillus* in the colon (*p* < 0.05). We hypothesized their potential roles as beneficial bacteria in the GBM obesity model. *Lactobacillus*, a widely applied probiotic, has demonstrated the ability to reduce inflammation in STZ-treated rats [57] and diabetic patients [58], and it can also regulate gut microbiota, thus, suppressing obesity [59]. Incorporating *Lactobacillus acidophilus* and *L. bulgaricus* into the diet of young guinea pigs improves intestinal health and consequently improves weight gain, reduces diarrhea and deaths, and normalizes the natural microbiota of the gastrointestinal tract [60]. We observed a declining trend in the relative abundance of Lactobacillus in the jejunum, ileum, and colon of the HFHCD group (*p* > 0.05). These findings suggest the pivotal role of these bacterial genera in HFHCD-induced obesity. Conclusively, the changes in microbial abundance at the phylum level imply the influence of obesity on the species composition at the genus level in GBM, with a more pronounced impact on the colon than the jejunum and ileum. The results suggest a dysbiosis of the intestinal microbiota in the obesity model of GBM.

LEfSe analysis is employed to identify, across phylum, class, order, family, and genus levels, critical gut microbiota that differentiate between the ND group and the HFHCD group. Sixteen distinct entities were identified, all originating from the colon, further accentuating the more pronounced impact of obesity on the colon compared with the jejunum and ileum. The gut microbiota of the colon was the most diverse, possibly accounting for its greater susceptibility. Beneficial bacteria such as Spirochaetota, *Candidatus_Saccharimonas*, and Saccharimonadaceae were significantly enriched in the ND group within the colon. Wu et al. [61] observed a significant reduction in Spirochaetota abundance in diarrheal cattle. In a mouse model of acute colitis treated with egg white peptide, the relative abundance of *Candidatus_Saccharimonas* increased noticeably, suggesting a role in immune regulation and intestinal repair [62]. Meanwhile, Saccharimonadaceae showed a significant decrease in the feces of hyperlipidemic mice [63]. Our research aligned with these findings. Butyricicoccaceae [64] and *Dorea* [65] were significantly enriched in the colon of the HFHCD group, and they are both gut bacteria that produce butyrate. Butyrate, categorized as a short-chain fatty acid (SCFA), primarily fuels colon cells [66]. While SCFAs at elevated levels are often deemed health-beneficia [67], contrasting research indicates that SCFA overproduction might contribute to obesity through enhanced energy storage. This phenomenon could stem from the heightened energy availability in the colon due to excessive SCFA production, potentially escalating obesity and subsequent weight gain. Notably, higher fecal SCFA concentrations have been observed in obese individuals relative to their lean counterparts [68,69,70], a discovery that resonates with our study’s outcomes. On the other hand, excessive butyrate concentrations can inhibit intestinal stem cell division by increasing butyrate in the intestinal crypt, thereby impeding intestinal self-repair post-injury [71]. Therefore, we speculated that in colon of the GBM obesity model, there was an excessive increase in the abundance of butyrate-producing bacteria, negatively impacting intestinal health. In this study, we identify gut microbiota potentially involved in the onset and progression of pig obesity, which appear to participate in the process of obesity induced by HFHCD. However, further research is needed to understand the deeper mechanisms underlying the influence of these gut microbiota on obesity. In summary, obesity affects the composition and diversity of gut microbiota in GBM, leading to the imbalance of gut microbiota and intestinal dysfunction.

Correlation analysis serves as an efficacious approach to bridging multiple omics outcomes [72]. In the differential microbiota discovered in this study, limited evidence exists regarding their correlation with inflammatory responses and immune interactions. Given the identification of DEGs associated with inflammatory responses and immune, we conducted correlation analysis to explore potential connections between the microbiota identified in this study and the DEGs. In the results of the correlation analysis, variations in the expression of *COL6A6* correlate positively with changes in abundance for *Solibacillus*, *norank_f__Saccharimonadaceae*, and *Candidatus_Saccharimonas*. In a similar vein, fluctuations in *NMI* expression correlate positively with abundance shifts in *unclassified_f__Butyricicoccaceae*. Identification of these microbial communities align with genus-level differential and LEfSe analytical outcomes. As previously mentioned, *COL6A6* functions as an inflammation suppressor gene, while *Solibacillus*, *norank_f__Saccharimonadaceae*, and *Candidatus_Saccharimonas* are deemed beneficial bacteria. Hence, *COL6A6* is likely to engage in interactions with these three microbial communities, collectively mitigating inflammatory responses and rectifying microbiota imbalance. In contrast, *NMI* acts as a pro-inflammatory gene, and *unclassified_f__Butyricicoccaceae* can potentially impede post-damage intestinal self-repair. Consequently, these two factors may collaboratively instigate inflammatory reactions and contribute to microbial imbalance through their interactions. Furthermore, the correlation results between the gut microbiota and physiological and biochemical indicators showed that *Solibacillus*, *norank_f__Saccharimonadaceae*, and *Candidatus_Saccharimonas* were negatively correlated with the pro-inflammatory cytokine IL-6 and blood-lipid-related indicators TG, TC, and FFA. Conversely, *unclassified_f__Butyricicoccaceae* correlated positively with these indicators. This provides additional evidence supporting the suppressive effects of *Solibacillus*, *norank_f__Saccharimonadaceae*, and *Candidatus_Saccharimonas* on inflammation and obesity, along with the promotional effect of *unclassified_f__Butyricicoccaceae*. In summary, by synthesizing insights from computational transcriptomics, microbial diversity analyses, and physiological as well as biochemical indicators, we infer that *Solibacillus*, *norank_f__Saccharimonadaceae*, *Candidatus_Saccharimonas*, and *unclassified_f__Butyricicoccaceae* are likely key microbiota implicated in inflammation and lipid metabolism. These communities may potentially interact with genes, including *COL6A6* and *NMI*, thereby influencing the onset and progression of obesity.

## 5. Conclusions

The results of this study indicate that HFHCD may play a role in the onset and progression of obesity through the enhancement of inflammatory responses, disruption of immune balance, and concurrent induction of gut microbiota imbalance and intestinal dysfunction in GBM. *COL6A6*, *NMI*, *Solibacillus*, *norank_f__Saccharimonadaceae*, *Candidatus_Saccharimonas*, and *unclassified_f__Butyricicoccaceae* may represent novel discoveries linked to obesity.

## Figures and Tables

**Figure 1 microorganisms-12-00369-f001:**
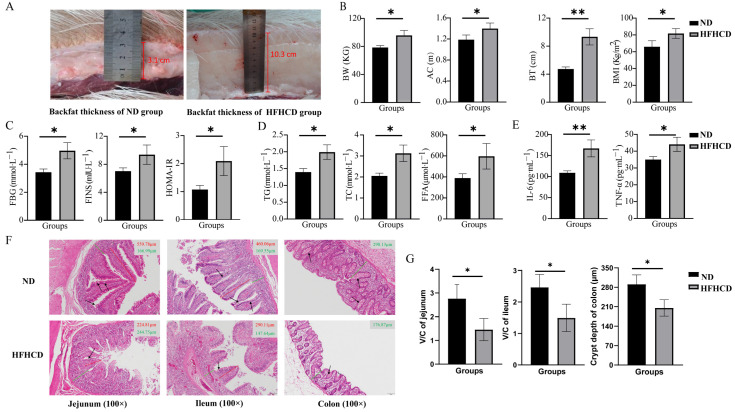
Construction of an obesity model in Guangxi Bama mini pigs and changes in their phenotypic characteristics. (**A**) Photographs indicating the measurement of the backfat thickness (BT). (**B**) The values of body weight (BW), abdominal circumference (AC), BT, and BMI. (**C**) The levels of fasting blood glucose (FBG), fasting insulin (FINS), and HOMA-IR. (**D**) The levels of triglycerides (TG), total cholesterol (TC), and free fatty acids (FFA). (**E**) The levels of interleukin-6 (IL-6) and tumor necrosis factor-alpha (TNF-α). (**F**) Images of the H&E-stained paraffin sections of jejunum, ileum, and colon (the red solid line representing the length of intestinal villi, the green solid line representing the height of the crypt, and the black arrow pointing to the goblet cells; 100×). (**G**) The V/C ratio in the jejunum and ileum and statistical analysis of the crypt depth in the colon based on the results of H&E staining. * *p* < 0.05 vs. ND group and ** *p* < 0.01 vs. ND group using Student’s *t*-test. Results are expressed as mean ± standard deviation $\left(\bar{x}\;\pm \;s,\;n=5\right).$ The figures were generated using GraphPad Prism 9.5.1.

**Figure 2 microorganisms-12-00369-f002:**
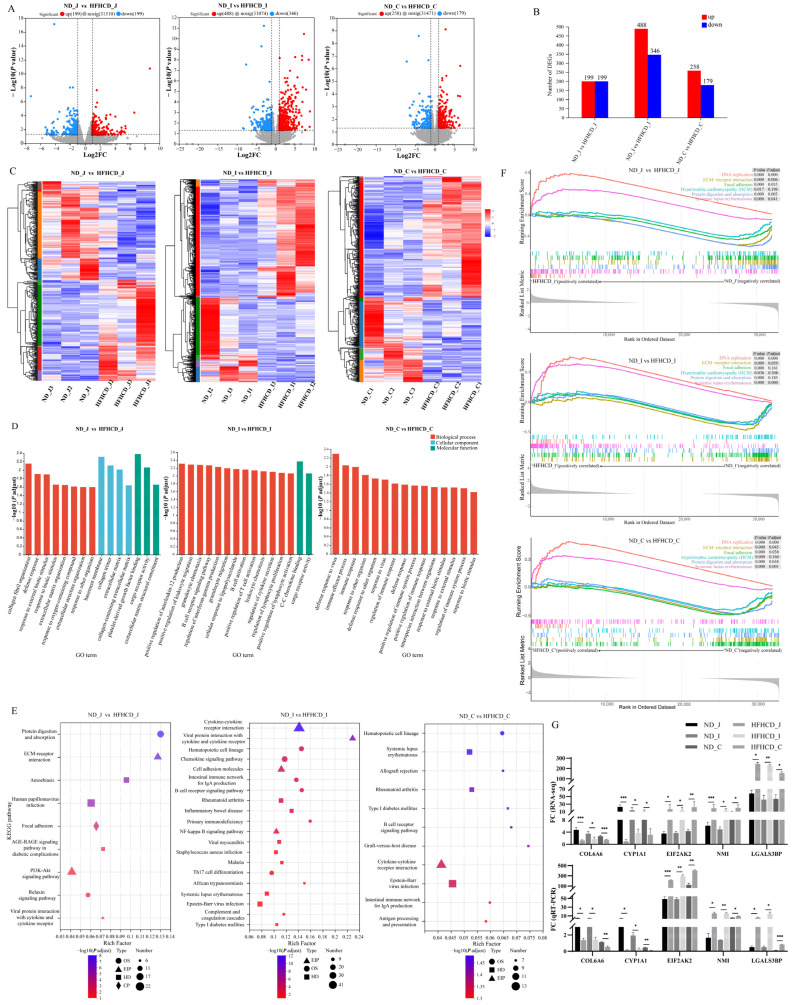
Identification, validation, and functional enrichment analysis of the differentially expressed genes (DEGs). (**A**) DEGs of jejunum, ileum, and colon in the HFHCD/ND gene set. The red dots indicate up-regulations of gene expression in the HFHCD group compared with the ND group, while the blue dots signify down-regulations. (**B**) Numbers of DEGs of jejunum, ileum, and colon in the HFHCD/ND gene set. The red rectangles represent up-regulation, while the blue ones indicate down = regulation. (**C**) Two-way hierarchical clustering heatmaps of up-regulated and down-regulated DEGs of jejunum, ileum, and colon in the HFHCD/ND gene set. Each column corresponds to a sample, while each row denotes a specific gene. The color coding reflects the normalized expression levels of the genes across the samples; red signifies higher expression of the gene in that sample, and blue indicates lower expression. (**D**) The enrichment maps of the GO biological function. (**E**) The KEGG pathway analysis. (**F**) The plot of the GSEA enrichment analysis demonstrates the enrichment patterns shared among the three intestinal segments. (**G**) qRT-PCR-based mRNA expression levels. * *p* < 0.05 vs. ND group, ** *p* < 0.01 vs. ND group, *** *p* < 0.001 vs. ND group using Student’s *t*-test. Results were expressed as mean ± standard deviation $\left(\bar{x}\;\pm \;s,\;n=3\right)$. ND_J, ND_I, and ND_C represent the jejunum, ileum, and colon of the ND group, respectively. HFHCD_J, HFHCD_I, and HFHCD_C represent the jejunum, ileum, and colon of the HFHCD group, respectively.

**Figure 3 microorganisms-12-00369-f003:**
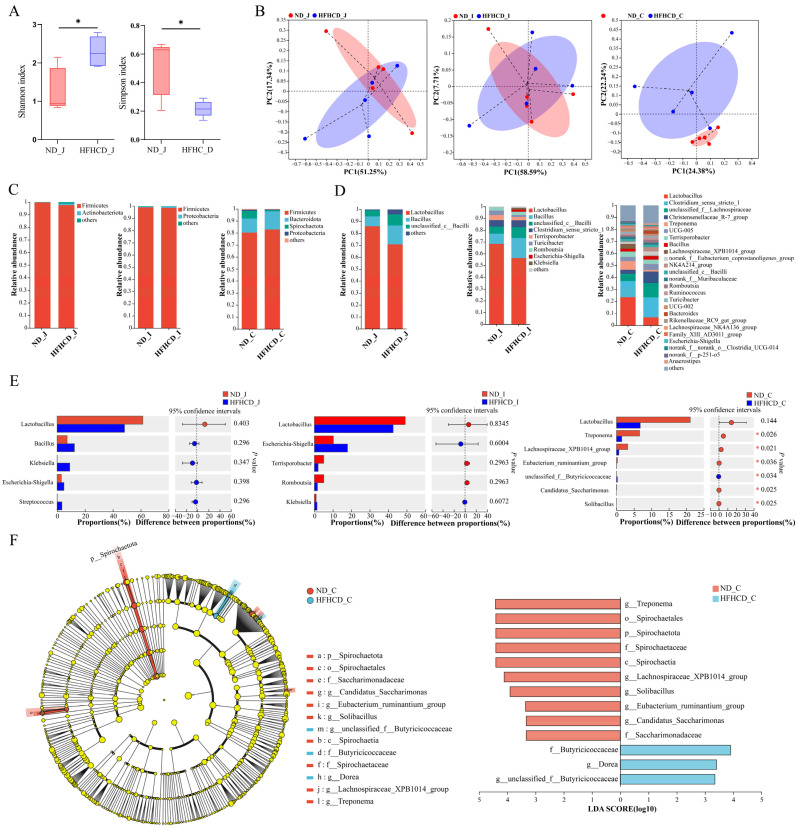
The composition and changes of gut microbiota in the GBM obesity model. (**A**) Simpson index and Shannon index of the jejunum. (**B**) PCoAs at the ASV level. Each point represents an individual sample. (**C**,**D**) Bar charts showing the species composition of the jejunum, ileum, and colon at the phylum (**C**) and genus (**D**) level. (**E**) Bar plots of the jejunum, ileum, and colon at the genus level. (**F**) Cladogram and column diagram of the colon. LEfSe analysis, LDA score > 2, *p* < 0.05. * *p* < 0.05 vs. ND group using Student’s *t* test (**A**) and Wilcoxon rank-sum test (**E**,**F**). Results are expressed as mean ± standard deviation $\left(\bar{x}\;\pm \;s,\;n=5\right)$. ND_J, ND_I, and ND_C represent the jejunum, ileum, and colon, respectively, of the ND group, and HFHCD_J, HFHCD_I, and HFHCD_C represent the jejunum, ileum, and colon, respectively, of the HFHCD group.

**Figure 4 microorganisms-12-00369-f004:**
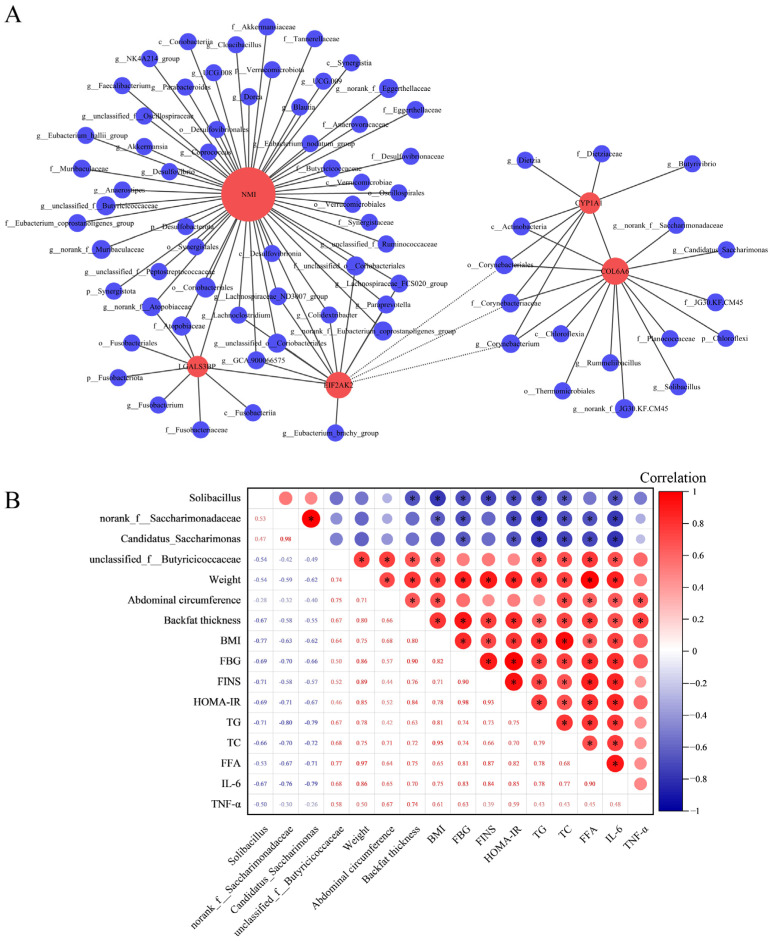
Integrated analysis of microbial diversity with transcriptomics and physiological biochemical indicators. (**A**) Correlation analysis of DEGs with microbial communities at the phylum, class, order, family, and genus levels. In the graphical representation, red circles denote gene names, and blue circles correspond to names of microbial communities. The diameter of these circles indicates the quantity of association pairs. Positive correlations are depicted with solid lines, whereas negative correlations are represented by dashed lines. Correlation coefficient absolute value > 0.5 and *p* adjust < 0.05. (**B**) Correlation analysis of gut microbiota and physiological biochemical indicators. In the upper right part of the matrix, the correlation index is indicated by the size and color of circles in the matrix’s upper right section, with numerical values displayed in the lower left corner. The red signifies a positive correlation, and the blue indicates a negative correlation. * *p* < 0.05.

**Table 1 microorganisms-12-00369-t001:** Standard of nutritional ingredients in pig feed.

Nutritional Ingredient	Guaranteed Analysis (%)
Crude protein	≥16.0
Lysine	≥0.8
Crude fiber	≤6.0
Crude ash	≤7.0
Calcium	0.6~1.2
Total phosphorus	0.4~1.0
Sodium chloride	0.2~0.8
Water	≤13.0

## Data Availability

The omics data from this study were deposited in the NCBI (https://www.ncbi.nlm.nih.gov, accessed on 6 February 2024). The BioProject accession number for the transcriptomic sequencing is PRJNA1021939, and for the 16S rRNA sequencing is PRJNA1047915.

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
