# Peer review of "Integrated Analysis of the Transcriptome and Microbial Diversity in the Intestine of Miniature Pig Obesity Model"

_microorganisms, 2024, doi:10.3390/microorganisms12020369_

Round 1

Reviewer 1 Report

Comments and Suggestions for Authors

In this paper, Authors analyze a miniature pig as a potential model for obesity studies. In particular, they observe and identify differentially expressed genes in pigs under normal and high fat-high carbohydrate diets. In addition, integration of data with analyses of microbial diversity show positive link between some microbes and changes in gene expression.

The manuscript appears well written and sound, however, some aspects should be addressed:

1)    Adaptability(?) and genetic stability of the Guangxi Bama miniature pigs considered for the analyses deserve a more complete picture;

2)    The number of considered individuals is not certainly high;

3)    Why only females are considered? I think that a robust model to be extended to the human being should be based on both sexes and on a higher number of samples;

4)    Do we have an idea whether intestinal tissues inflammation is determined or followed by the development of the different bacterial populations?

Author Response

Dear Editor and Reviewers,

First and foremost, I would like to express my sincere gratitude for the time and effort spent reviewing my manuscript titled "Integrated Analysis of the Transcriptome and Microbial Diversity in the Intestine of Miniature Pig Obesity Model" and for the constructive comments provided. These insights have been invaluable in enhancing the quality of my work.

Following the comments received, I have meticulously addressed each point raised. Below, I provide a summary of the main issues highlighted and the corresponding revisions made to the manuscript. Please find the detailed responses below and the highlighted in the resubmitted files.

Response to Reviewer #1:

  1. Comment 1: Adaptability(?) and genetic stability of the Guangxi Bama miniature pigs considered for the analyses deserve a more complete picture.

Reply 1: Dear reviewer, we appreciate your perceptive insights and considerations regarding the adaptability and genetic stability of the Guangxi Bama miniature pigs. In response, we have enriched our article with a comprehensive discussion that further explores the potential of Guangxi Bama miniature pigs as a robust and genetically stable model in disease research. They are in lines 45-50 of the manuscript. For more information on the breeding history and genetic characteristics of the Bama miniature pig inbred line, please check out our 2019 iScience article (Zhang, L.; Huang, Y.; Wang, M.; Guo, Y.; Liang, J.; Yang, X.; Qi, W.; Wu, Y.; Si, J.; Zhu, S.; et al. Development and Genome Sequencing of a Laboratory-Inbred Miniature Pig Facilitates Study of Human Diabetic Diseases. IScience 2019, 19, 162-176, doi: 10.1016 / j.i sci. 2019.07.025).

“In particular, Guangxi Bama miniature pig (GBM) possesses characteristics such as small body size, ease of rearing, strong adaptability, and genetic stability, making them an ideal experimental animal model for studying diet-related cardiovascular metabolic diseases [4–6]. GBM is a breeding line that has been inbred for over 30 years (since 1987), featuring a comparatively stable genetic background [4,7]. This makes it a critical animal model for researching human diseases.”

  1. Comment 2: The number of considered individuals is not certainly high.

Reply 2: Thank you for imparting your esteemed perspectives. Pertaining to this inquiry, I would like to direct you to "Reply 1" for a detailed response. The Guangxi Bama miniature pigs employed in our study are a meticulously cultivated inbred strain, established by our dedicated team. GBM has clear genetic background (without foreign gene flow influx), stable characters and high homozygosity (inbreeding coefficient of inbred line F19: 0.9825), they are distinguished by their consistent genetic attributes and minimal individual variances. In the realm of experimental animal models, these pigs, notably, require a smaller cohort to yield dependable outcomes when compared to other miniature pig breeds. As such, we are confident that the data garnered from groups of merely five individuals in our study offers a high degree of accuracy, close to the number of individuals we used in our previous study. Moving forward, we are poised to broaden our research scope by augmenting the number of subjects in each group, thereby enriching the robustness of our findings.

  1. Comment 3: Why only females are considered? I think that a robust model to be extended to the human being should be based on both sexes and on a higher number of samples.

Reply 3: Your insights on the matter of gender in scientific research are immensely appreciated, as this is indeed a topic of profound specialization. In our earlier endeavors, we utilized male Guangxi Bama miniature pigs as models for T2DM studies, yielding favorable outcomes. In our current endeavor, aiming to enhance the strategic application of GBM in models of metabolic diseases, we have chosen to incorporate female specimens. The contribution of female animals to the study of metabolic disorders is significant and enduring. As such, female GBMs are exceptionally suited for use as experimental subjects in our research. This topic is further explored in lines 334-337 of our manuscript. Your valuable feedback has inspired us to contemplate future research that compares disease progression across different genders, a subject richly deserving of in-depth exploration.

“Our preceding studies have established GBM as one of the optimal candidate for developing a type 2 diabetes (T2DM) model, yet the underlying molecular mechanisms remain elusive [4,25]. The utilization of female miniature pig models in the studies of metabolic diseases has yielded gratifying outcomes [26,27].”

  1. Comment 4: Do we have an idea whether intestinal tissues inflammation is determined or followed by the development of the different bacterial populations?

Reply 4: Thank you for highlighting the intricate aspect of "cause or effect" in our study. We delve into the nuanced interplay between gut microbiota dysbiosis and intestinal inflammation, particularly in the context of obesity induced by HFHCD. This relationship is characteristically bidirectional; it is not merely a case of gut microbiota dysbiosis leading to intestinal inflammation, but also the reverse, where intestinal inflammation can precipitate changes in the gut microbiota composition. This dual influence suggests a complex, intertwined relationship between gut microbiota dysbiosis and intestinal inflammation, one that potentially spirals into a harmful cycle: dysbiosis triggers inflammation, which in turn exacerbates the microbiota imbalance. Our manuscript delves deeper into this topic, offering an extensive discussion in lines 406-421.

“Additionally, emerging research highlights a profound interconnection between gut microbiota dysbiosis and intestinal inflammation. Under normal conditions, there exists a symbiotic equilibrium between the gut microbiota and its host. This harmony is pivotal in maintaining the integrity of the intestinal mucosa, modulating the immune system, and generating metabolites that are beneficial for the host. However, when this delicate balance is perturbed, it may result in an increase in pathogenic bacteria and a decrease in beneficial ones. Such an imbalance can initiate an immune response in the intestinal mucosa, leading to inflammation. This inflammatory response can further compromise the intestinal mucosal barrier, thereby intensifying the dysbiosis [45]. Moreover, the intestinal inflammation itself can catalyze changes in the gut environment, influencing the microbiota composition. The mucosal damage and altered immune responses brought on by inflammation can foster an environment that favors certain pathogens, while being hostile to beneficial bacteria, thus exacerbating the dysbiosis [46]. Consequently, the dynamics between gut microbiota dysbiosis and intestinal inflammation are intertwined, potentially spiraling into a deleterious cycle: microbiota dysbiosis triggers intestinal inflammation, which in turn exacerbates the disruption of the microbiota equilibrium.”

In addition to addressing the major concerns, we have also added an acknowledgment section to express our gratitude to Dr. Ting Yang (Guangxi University) for her contributions to our manuscript. They are in lines 569-572 of the manuscript.

Acknowledgements: We thank Dr. Ting Yang (Guangxi University) for the meticulous revision and enhancement of our manuscript. Her expertise and attention to detail significantly elevated the quality of our document, enhancing its fluency, clarity, and expression. These contributions have been invaluable to our work.”

I hope these revisions meet the expectations of the reviewers, and I look forward to any further guidance or suggestions you might have. Thank you for your continued interest in and support of my work.

In closing, I would like to extend my thanks once again to you and the reviewers for your insightful comments and the opportunity to improve my manuscript.

Sincerely,

Wenjing Qi

Affiliation: College of Animal Science and Technology, Guangxi University, Nanning 530004, China

Postal address: Guangxi University, No. 100, University East Road, Xixiangtang District, Nanning, P.R. China

Phone number: +(86) 15177136967

Email address: [email protected]

Reviewer 2 Report

Comments and Suggestions for Authors

Brief Summary

The article describes the impact of high fat high carbohydrate diet on Guangxi Bama miniature pig model, associating microbiota investigation and transcriptomic to identify Key biomarkers. The article is written in very good English, and the material and methods used seem adequate as far as I can judge. The presentation of the many results needs to be revised slightly, as well as a few points for discussion.

General Concept

My main criticism is about the purpose of the GBM model, which seems very attractive at first glance. However, I am concerned about some results obtained with the help of this model.

In the jejunum of HFHCD group, the Shannon index was elevated, and the Simpson index reduced. This denoted an enhanced alpha diversity. In most studies in humans, the results are reversed, namely a significantly lower alpha diversity (Shannon index) in obese versus non-obese adults. Could you comment on that in the discussion?

You also mention in the discussion that butyrate production would be rather negative, especially for mucosal parameters, although there are many papers reporting a protective Role of Butyrate against Obesity. Same comments as above could you review this point in the discussion ?

Specific comments referring to line numbers,

Line 64 : can you justify the choice of female GBM pigs ?

Line 65 : five pigs per group is very few, but apparently variability was low for most of the parameters tested.

Line 66 could you give a table with the composition of the diet ?

Line 90 : Can you mention if this study was supervised by an ethics committee?

Line 188 : figure 2 a-c Can you present these figures (especially 2-c) in a more readable way? As it stands, it's impossible to read !

Line 387-389 : see General concept comments

Line 439-444 : see General concept comments  

Author Response

Dear Editor and Reviewers,

First and foremost, I would like to express my sincere gratitude for the time and effort spent reviewing my manuscript titled "Integrated Analysis of the Transcriptome and Microbial Diversity in the Intestine of Miniature Pig Obesity Model" and for the constructive comments provided. These insights have been invaluable in enhancing the quality of my work.

Following the comments received, I have meticulously addressed each point raised. Below, I provide a summary of the main issues highlighted and the corresponding revisions made to the manuscript. Please find the detailed responses below and the highlighted in the resubmitted files.

Response to Reviewer #2:

  1. Comment 1: My main criticism is about the purpose of the GBM model, which seems very attractive at first glance. However, I am concerned about some results obtained with the help of this model.

In the jejunum of HFHCD group, the Shannon index was elevated, and the Simpson index reduced. This denoted an enhanced alpha diversity. In most studies in humans, the results are reversed, namely a significantly lower alpha diversity (Shannon index) in obese versus non-obese adults. Could you comment on that in the discussion?

Reply 1: Dear reviewer, thank you for your insightful comments and valuable suggestions on our manuscript. First, we meticulously refined our narrative surrounding the findings on alpha diversity to enhance clarity and depth. They are in lines 240-245 and 258 of the manuscript. The Shannon and Simpson indices are metrics used to assess diversity, where a higher Shannon index and a lower Simpson index indicate greater gut microbiota diversity. While some studies have demonstrated a decrease in the diversity of the gut microbiome in individuals suffering from obesity and conditions like T2DM, a number of studies present contrary evidence, challenging this widely accepted trend. In our study, in the HFHCD group, the ileum's Shannon index was significantly higher, and the Simpson index was significantly lower compared with the ND group. This observed pattern reveals an increased α-diversity in the jejunum of the HFHCD group, aligning with some of the prior research findings and corroborating our investigative results in the part of “community composition and variation”. In the ileum, 182 genera were common between both groups, alongside 108 unique to the ND group (Figure S1C) and 21 unique to the HFHCD group (Figure S1D). (They are in lines 276-278 of the manuscript.) This indicates a richer gut microbiota diversity in the HFHCD group, which includes some harmful bacteria. We discuss this point in our manuscript, detailed in lines 422-442.

240-245:

“Our analysis of the Sobs, ACE, Chao, Shannon, and Simpson indices revealed two signif-icant differences: in the HFHCD group, the jejunum's Shannon index was significantly higher, and the Simpson index was significantly lower compared with the ND group (Figure 3A). In both the ileum and colon, the diversity indices of the intestinal microbiota exhibited no significant variations between those in the ND and FHHCD groups.”

258:

“(A) Simpson index and Shannon index of the jejunum.”

276-278:

“In the ileum, 182 genera were common between both groups, alongside 108 unique to the ND group (Figure S1C) and 21 unique to the HFHCD group (Figure S1D).”

422-442:

“The Shannon index is directly proportional to biodiversity, while the Simpson index is inversely proportional to biodiversity [47]. While some studies have demonstrated a decrease in the diversity of the gut microbiome in individuals suffering from obesity and conditions like T2DM [48,49], a number of studies present contrary evidence, challenging this widely accepted trend. For example, this research indicates that the diversity of the gut microbiome in individuals with obesity is notably greater compared to those who are not obese.[50]. In comparison to non-diabetic individuals, those diagnosed with diabetes demonstrate enhanced diversity in the microbiome of their fecal samples [51]. Our study revealed that in the jejunum of the HFHCD group, the Shannon index was elevated, and the Simpson index was reduced. This indicated an elevated level of alpha diversity, which is consistent with some of the results documented in the previously referenced studies. This finding corresponded with the identification of a greater variety of specific bacterial genera in the jejunum. Notably, this group included harmful bacteria like Veillonella, capable of colonizing the intestines under inflammatory conditions [52]. The escalation in the presence of microbes implicated in inflammatory processes might have contributed to the augmented microbial diversity. The dynamics of immune and inflammatory responses emerge as critical areas of focus in obesity model research, underscoring the potential outcome of a reciprocal interplay between the host and gut microbiota during the development and progression of obesity. To sum up, when it comes to exploring the diversity of the gut microbiome in individuals with obesity, it is imperative that further detailed investigations are conducted to uncover the intricate mechanisms at play.”

  1. Comment 2: You also mention in the discussion that butyrate production would be rather negative, especially for mucosal parameters, although there are many papers reporting a protective Role of Butyrate against Obesity. Same comments as above could you review this point in the discussion?

Reply 2: Thank you for your insightful inquiry, demonstrating both depth of knowledge and specialized expertise. While SCFAs at elevated levels are often deemed health-beneficia [63], contrasting research indicates that SCFA overproduction might contribute to obesity through enhanced energy storage. On the other hand, Excessive butyrate concentrations can inhibit intestinal stem cell division by increasing butyrate in the intestinal crypt, thereby impeding intestinal self-repair post-injury. This observation leads us to hypothesize that an overabundance of butyrate-producing bacteria in the colon of the GBM obesity model may detrimentally impact intestinal health. These findings are elaborated upon in lines 490-503 of our manuscript.

“Butyricicoccaceae [64] and Dorea [65] were significantly enriched in the colon of the HFHCD group, and they are both gut bacteria that produce butyrate. Butyrate, categorized as a short-chain fatty acid (SCFA), primarily fuels colon cells [66]. While SCFAs at elevated levels are often deemed health-beneficia [67], contrasting research indicates that SCFA overproduction might contribute to obesity through enhanced energy storage. This phenomenon could stem from the heightened energy availability in the colon due to excessive SCFA production, potentially escalating obesity and subsequent weight gain. Notably, higher fecal SCFA concentrations have been observed in obese individuals relative to their lean counterparts [68–70], a discovery that resonates with our study's outcomes. On the other hand, Excessive butyrate concentrations can inhibit intestinal stem cell division by increasing butyrate in the intestinal crypt, thereby impeding intestinal self-repair post-injury [71]. Therefore, we speculated that in colon of the GBM obesity model, there was an excessive increase in the abundance of butyrate-producing bacteria, negatively impacting intestinal health.”

  1. Comment 3: Line 64: can you justify the choice of female GBM pigs?

Reply 3: We deeply appreciate your valuable insights regarding gender considerations in our research. In light of the promising results achieved with male Guangxi Bama miniature pigs in our previous studies focusing on T2DM, we have deliberately chosen female pigs for our current research. This decision is part of a strategic effort to enhance the application of Guangxi Bama miniature pigs as a model for metabolic diseases. Ensuring gender consistency is a critical aspect of our research methodology, as it contributes to the uniformity and reliability of the study's context. Looking ahead, we plan to expand our research scope to include comparative analyses of disease manifestations across different genders. We have elaborated on this topic in lines 334-337 of our manuscript.

“Our preceding studies have established GBM as one of the optimal candidate for developing a type 2 diabetes (T2DM) model, yet the underlying molecular mechanisms remain elusive [4,25]. The utilization of female miniature pig models in the study of metabolic diseases has yielded gratifying outcomes [26,27].”

  1. Comment 4: Line 65: five pigs per group is very few, but apparently variability was low for most of the parameters tested.

Reply 4: We are grateful for your focus on this particular aspect of our study. The Guangxi Bama miniature pigs utilized in our research are a closely inbred strain, meticulously developed by our team. GBM has clear genetic background (without foreign gene flow influx), stable characters and high homozygosity (inbreeding coefficient of inbred line F19: 0.9825), and are noted for their consistent genetic profile and minor individual variations. As models in experimental settings, these pigs generally require a smaller cohort to achieve results that are reliable and representative, especially when compared to other miniature pig breeds. Consequently, we are confident that the data derived from groups of five individuals in this study is robust and reflective of the broader population. Nonetheless, we recognize the importance of sample size in validating research findings and anticipate increasing the number of subjects in future studies. To underscore the significance of these considerations, we have devoted a detailed discussion in our manuscript to the adaptability and genetic consistency of Guangxi Bama miniature pigs as disease models, specifically outlined in lines 45-50. For more information on the breeding history and genetic characteristics of the Bama miniature pig inbred line, please check out our 2019 iScience article (Zhang, L.; Huang, Y.; Wang, M.; Guo, Y.; Liang, J.; Yang, X.; Qi, W.; Wu, Y.; Si, J.; Zhu, S.; et al. Development and Genome Sequencing of a Laboratory-Inbred Miniature Pig Facilitates Study of Human Diabetic Diseases. IScience 2019, 19, 162-176, doi: 10.1016 / j.i sci. 2019.07.025).

“In particular, Guangxi Bama miniature pig (GBM) possesses characteristics such as small body size, ease of rearing, strong adaptability, and genetic stability, making them an ideal experimental animal model for studying diet-related cardiovascular metabolic diseases [4–6]. GBM is a breeding line that has been inbred for over 30 years (since 1987), featuring a comparatively stable genetic background [4,7]. This makes it a critical animal model for researching human diseases.”

  1. Comment 5: Line 66 could you give a table with the composition of the diet?

Reply 5: Thank you for your suggestions regarding the manuscript. I have supplemented a table with the composition of the diet at lines 75-77 for this particular section.

“The standard of nutritional ingredients of the miniature pig feed are shown in Table 1.

Table 1 Standard of nutritional ingredients in pig feed

Nutritional ingredient

Guaranteed Analysis (%)

Crude protein

≥ 16.0

Lysine

≥ 0.8

Crude fibre

≤ 6.0

Crude ash

≤ 7.0

Calcium

0.6~1.2

Total phosphorus

0.4~1.0

Sodium chloride

0.2~0.8

Water

≤ 13.0

  1. Comment 6: Can you mention if this study was supervised by an ethics committee?

Reply 6:  Thank you for your reminder. This study has approved by the Ethics Committee of Guangxi University (GXU2018-006), and all the feeding, care, and experimental procedures of the experimental animals were strictly carried out in accordance with the Experimental Animal Management Regulations (amendment on March 1st, 2017, China). They are in lines 561-564 of the manuscript.

Institutional Review Board Statement: This study has approved by the Ethics Committee of Guangxi University (GXU2018-006), and all the feeding, care, and experimental procedures of the experimental animals were strictly carried out in accordance with the Experimental Animal Management Regulations (amendment on March 1st, 2017, China).”

  1. Comment 7: figure 2 a-c Can you present these figures (especially 2-c) in a more readable way? As it stands, it's impossible to read!

Reply 7: Thank you very much for your reminder. We have supplemented the text with explanations for "figure 2 A-C". They are in lines 175-179, and 203-211 of the manuscript.

175-179:

“In total, 1,669 DEGs were identified in the HFHCD/ND groups, including 398 in the jejunum, 834 in the ileum, and 437 in the colon (Figure 2A and B-C). The results from the enrichment analysis of DEGs revealed that, across all three intestinal segments, there was significant consistency within each group and distinct differentiation between the two groups (Figure 2C).”

203-211:

“(A) DEGs of jejunum, ileum and colon in the HFHCD/ND gene set. The red dots indicate upregulations of gene expression in the HFHCD group compared with the ND group, while the blue dots signify downregulations. (B) Numbers of DEGs of jejunum, ileum and colon in the HFHCD/ND gene set. The red rectangles represent upregulation, while the blue ones indicate downregulation. (C) Two-way hierarchical clustering heatmaps of upregulated and down-regulated DEGs of jejunum, ileum and colon in the HFHCD/ND gene set. Each column corresponds to a sample, while each row denotes a specific gene. The color coding reflects the normalized expression levels of the genes across the samples, with red signifies higher expression of the gene in that sample, and blue indicates lower expression.”

  1. Comment 8: Line 387-389: see General concept comments.

Reply 8: Thank you very much for your reminder. We have made additions to this section of the content and they are in lines 422-442 of the manuscript.

“The Shannon index is directly proportional to biodiversity, while the Simpson index is inversely proportional to biodiversity [47]. While some studies have demonstrated a de-crease in the diversity of the gut microbiome in individuals suffering from obesity and conditions like T2DM [48,49], a number of studies present contrary evidence, challenging this widely accepted trend. For example, this research indicates that the diversity of the gut microbiome in individuals with obesity is notably greater compared to those who are not obese.[50]. In comparison to non-diabetic individuals, those diagnosed with diabetes demonstrate enhanced diversity in the microbiome of their fecal samples [51]. Our study revealed that in the jejunum of the HFHCD group, the Shannon index was elevated, and the Simpson index was reduced. This indicated an elevated level of alpha diversity, which is consistent with some of the results documented in the previously referenced studies. This finding corresponded with the identification of a greater variety of specific bacterial genera in the jejunum. Notably, this group included harmful bacteria like Veillonella, capable of colonizing the intestines under inflammatory conditions [52]. The escalation in the presence of microbes implicated in inflammatory processes might have contributed to the augmented microbial diversity. The dynamics of immune and inflammatory responses emerge as critical areas of focus in obesity model research, underscoring the potential outcome of a reciprocal interplay between the host and gut microbiota during the development and progression of obesity. To sum up, when it comes to exploring the diversity of the gut microbiome in individuals with obesity, it is imperative that further detailed investigations are conducted to uncover the intricate mechanisms at play.”

  1. Comment 9: Line 439-444: see General concept comments

Reply 9: Thank you very much for your reminder. We have made additions to this section of the content and they are in lines 490-503 of the manuscript.

“Butyricicoccaceae [64] and Dorea [65] were significantly enriched in the colon of the HFHCD group, and they are both gut bacteria that produce butyrate. Butyrate, categorized as a short-chain fatty acid (SCFA), primarily fuels colon cells [66]. While SCFAs at elevated levels are often deemed health-beneficia [67], contrasting research indicates that SCFA overproduction might contribute to obesity through enhanced energy storage. This phenomenon could stem from the heightened energy availability in the colon due to excessive SCFA production, potentially escalating obesity and subsequent weight gain. Notably, higher fecal SCFA concentrations have been observed in obese individuals relative to their lean counterparts [68–70], a discovery that resonates with our study's outcomes. On the other hand, Excessive butyrate concentrations can inhibit intestinal stem cell division by increasing butyrate in the intestinal crypt, thereby impeding intestinal self-repair post-injury [71]. Therefore, we speculated that in colon of the GBM obesity model, there was an excessive increase in the abundance of butyrate-producing bacteria, negatively impacting intestinal health.”

In addition to addressing the major concerns, we have also added an acknowledgment section to express our gratitude to Dr. Ting Yang (Guangxi University) for her contributions to our manuscript. They are in lines 569-572 of the manuscript.

Acknowledgements: We thank Dr. Ting Yang (Guangxi University) for the meticulous revision and enhancement of our manuscript. Her expertise and attention to detail significantly elevated the quality of our document, enhancing its fluency, clarity, and expression. These contributions have been invaluable to our work.”

I hope these revisions meet the expectations of the reviewers, and I look forward to any further guidance or suggestions you might have. Thank you for your continued interest in and support of my work.

In closing, I would like to extend my thanks once again to you and the reviewers for your insightful comments and the opportunity to improve my manuscript.

Sincerely,

Wenjing Qi

Affiliation: College of Animal Science and Technology, Guangxi University, Nanning 530004, China

Postal address: Guangxi University, No. 100, University East Road, Xixiangtang District, Nanning, P.R. China

Phone number: +(86) 15177136967

Email address: [email protected]
